# Does Dynamic Anterior Plate Fixation Provide Adequate Stability for Traumatic Subaxial Cervical Spine Fractures at Mid-Term Follow-Up?

**DOI:** 10.3390/jcm10061185

**Published:** 2021-03-12

**Authors:** Siegmund Lang, Carsten Neumann, Lasse Fiedler, Volker Alt, Markus Loibl, Maximilian Kerschbaum

**Affiliations:** 1Department of Trauma Surgery, University Medical Center Regensburg, Franz-Josef-Strauss-Allee 11, 93053 Regensburg, Germany; siegmund.lang@ukr.de (S.L.); carsten.neumann@ukr.de (C.N.); fiedler.lasse@gmail.com (L.F.); volker.alt@ukr.de (V.A.); markus.loibl@kws.ch (M.L.); 2Department of Spine Surgery, Schulthess Clinic Zurich, Lenghalde 2, 8008 Zurich, Switzerland

**Keywords:** cervical spine trauma, severe injury, anterior cervical plate, dynamic plate concept, radiological follow-up, fusion, patient reported outcome measurement

## Abstract

Background: It remains questionable if the treatment of cervical fractures with dynamic plates in trauma surgery provides adequate stability for unstable fractures with disco-ligamentous injuries. The primary goal of this study was to assess the radiological and mid-term patient-reported outcome of traumatic subaxial cervical fractures treated with different plate systems. Patients and Methods: Patients, treated with anterior cervical discectomy and fusion (ACDF) between 2001 and 2015, using either a dynamic plate (DP: Mambo™, Ulrich, Germany) or a rigid locking plate (RP: CSLP™, Depuy Synthes, USA), were identified. For radiological evaluation, the sagittal alignment, the sagittal anterior translation and the bony consolidation were evaluated. After at least two years, the patient-reported outcome measures (PROM) were evaluated using the German Short-Form 36 (SF-36), Neck Disability Index (NDI) and the EuroQol in 5 Dimensions (EQ-5D) scores. Results: 33 patients met the inclusion criteria (DP: 13; RP:20). Twenty-six patients suffered from AO Type B or C fractures. Both the sagittal alignment and the sagittal translation could be sufficiently improved in both groups (*p* ≥ 0.05). No significant loss of reduction could be observed at the follow-up in both groups (*p* ≥ 0.05). Bony consolidation could be observed in 30 patients (DP: 12/13 (92%); RP: 18/20 (90%); (*p* ≥ 0.05)). In 20 patients, PROMs could be evaluated (follow-up: 71.2 ± 25.5 months). The whole cohort showed satisfactory PROM results (EQ-5D: 72.0 ± 4.9; SF-36 PCS: 41.9 ± 16.2, MCS: 45.4 ± 14.9; NDI: 11.0 ± 9.1). without significant differences between the DP and RP group (*p* ≥ 0.05) Conclusion: The dynamic plate concept provides enough stability without a difference in fusion rates in comparison to rigid locking plates in a population that mostly suffered fragile fractures.

## 1. Introduction

In degenerative spine surgery as well as in trauma surgery, anterior cervical discectomy and fusion (ACDF), combining a cage or bone-graft with an anterior plate, is an established treatment of various cervical pathologies [1,2]. Anterior cervical fixation procedures with different plate- and screw-systems with the goal of solid fusion of the segment have evolved since the 1960s [3,4]. Implant-associated complications such as pseudarthrosis, kyphotic segmental deformations, graft dislocation, and screw and plate displacements were reported [5,6]. Consequently, dynamic plates were introduced, allowing a greater axial load on the bone graft which might contribute to solid fusion. Few meta-analyses have evaluated the benefits of dynamic plate systems compared to static systems; however, most of the clinical studies were conducted for degenerative indications [7,8]. The results regarding a benefit of dynamic plates on fusion rates are controversial, with a trend to favoring the dynamic concept, especially for multi-segmental fusions [9]. In terms of clinical outcome, no clear advantages of dynamic plates versus rigid plates have been reported [8,10]. The literature on the use of dynamic plate systems in the treatment of cervical disco-ligamentous injuries and unstable fractures is sparse. In contrast to degenerative pathologies, traumatic pathologies of the cervical spine inherit remarkable instability, especially when the posterior spinal elements are injured [11]. For this reason, it is questionable if a dynamic plate system provides adequate stability to ensure stable fracture reduction and consolidation over time. 

The current study aims to analyze the radiological and mid-term patient-reported outcome measures after ACDF of cervical disco-ligamentous injuries and cervical spine fractures, comparing a dynamic plate system (Mambo™, Ulrich Medical) and a rigid plate system (CSLP™, Depuy Synthes) in a cohort of 33 trauma patients. 

## 2. Materials and Methods

### 2.1. Inclusion and Exclusion Criteria

Patients with traumatic subaxial, cervical spinal fractures, who received a plate fixation by an anterolateral approach either with a dynamic (DP) or a rigid plate (RP) in our trauma department (level 1 trauma center) between 01/2001 and 01/2015, were identified for this study. All surgeries were carried out by one senior spine surgeon (C.N.) or under his supervision. The indications for anterior plating alone were made according to the individual patient situation, fracture classification and the surgeon’s experience. Patients younger than 18 years, as well as patients with a pathologic fracture (including osteoporotic fractures), ankylosing spondylitis, incomplete radiological follow-up (minimum one year) or complete paralysis were excluded. Next to the patient related data (sex, age), the injury mechanism, Injury Severity Score (ISS), fracture type and the treatment details (surgical strategy; adverse events) were assessed. Patient-reported outcome measures (PROMs) were evaluated after at least two years.

### 2.2. Radiographic Assessment

All included patients received pre- and postoperative CT scans and anterior-posterior and lateral X-rays of the cervical spine at follow-up. The height of the anterior and the posterior wall (Figure 1A) and the sagittal translation (Figure 1B) was measured. For the assessment of segmental fusion, the absence of radiolucencies, the absence of bone sclerosis and evidence of bridging trabecular bone within the fusion area were evaluated in the lateral X-rays at follow-up. The mono-segmental (mEA) and bisegemental endplate angles (bEA) were measured as shown in Figure 1A depending on the number of segments fused [12]. Negative values indicate lordosis, and positive values kyphosis. All measurements were performed digitally by the same, trained physician, using the software package OsiriX MD (Pixmeo, Bernex, Switzerland).

### 2.3. Patient-Reported Outcome Measures (PROM)

The German Short-Form 36 (SF-36) [13], Neck Disability Index (NDI) [14] and EuroQol in 5 Dimensions (EQ-5D) were used to assess PROM and quality of life. Patients were contacted by telephone and asked for their participation in the study, starting two years after surgery, prior to sending the PROM questionnaires along with the written consent form. If patients were not reachable by phone, forms were sent to the last registered address. As a reference, normative data from Germany were used on SF-36 [15] and EQ-5D [16]. For the SF-36, the physical components summary (PCS) and mental components summary (MCS) were calculated as described by Ellert et al. [15].

### 2.4. Statistics

Statistical analysis was carried out using SPSS software version 24 (SPSS Inc, Chicago, IL, USA). Variables were tested for normal distribution with the Shapiro–Wilk test. Mann–Whitney U and Kruskal–Wallis tests were performed to compare categorical variables; the independent t-test was used to compare continuous variables. Due to the low number of failures, statistical evaluation is limited to bivariate analysis. *p*-values < 0.05 were considered significant. Data are presented as mean ± standard deviation (SD) for continuous variables and as absolute and relative frequencies for categorical data. 

## 3. Results

### 3.1. Demographics

Thirty-three patients (8 females, 25 males, mean age 48.2 ± 22.2 years) met the inclusion criteria including a complete radiological follow-up (29.8 ± 24.3 months). Twenty patients (6 females, 14 males, mean age 48.3 ± 22.0 years) completed the PROM questionnaires after a mean follow-up of 71.2 ± 25.5 months. Loss to follow-up was 39%. There were no statistically significant differences in demographic data and the ISS of patients that were lost to follow-up (mean age 48.2 ± 23.5 years; ISS: 20 ± 16 points; 2 females, 11 males) and the patients that completed the follow-up (mean age 48.3 ± 22.0 years; ISS: 20 ± 15 points; 6 females, 14 males) (all *p* ≥ 0.05). The flowchart in Figure 2 displays the patient enrollment.

### 3.2. Trauma Mechanism and Fracture Classification

Twelve patients (36%) suffered traffic accidents, 4 (12%) a fall from more than 3 m, and 15 (46%) from less than 3 m. Two patients (6%) had an unclassified cause of trauma. Fourteen patients (42%) suffered from multiple injuries with an Injury Severity Score (ISS) ≥ 16. Out of those severely injured patients, seven patients completed the PROM questionnaires. The segment C6/7 was the most frequently affected segment (*n* = 12, 36%). AO Type A fractures were seen in seven patients (Type AO A3 in *n* = 5 and Type AO A4 in *n* = 2). Fractures were classified as AO Type B and C fractures in 26 cases (79%) and there was no significant difference in fracture type distribution between the two groups (Table 1). Preoperatively, 12 patients (36%) suffered neurological impairment with motor deficits in 6 cases (18%) and sensory disturbances in 6 cases (18%). A motor deficit persisted postoperatively in 4 cases (12%) and a sensory deficit in 2 cases (6%). 

### 3.3. Treatment Strategy and Complications

Closed reduction was attempted in all patients. If this failed, an anterior open reduction was performed. Closed or open reduction via an anterior approach was successful in all included patients. A dynamic plate was used in 13 patients (39%). Rigid plate fixation was performed in 20 patients (61%) (Table 1). 23 patients (70%) were treated with a one-level ACDF, eight (24%) with a two-level fixation and two (6%) with a three-level ACDF. Looking at the three-level fixations, one dynamic and one rigid plate concept were applied. Due to this dichotomous distribution of three-level ACDFs, those two cases were not included in the statistical analysis of changes in the endplate angles or anterior translation, but in all other evaluations. In 27 cases (82%), an iliac crest bone graft (ICBG) and in 6 cases (18%), a titanium cage with local bone graft were used to provide anterior support and achieve fusion. The bone crests were harvested exclusively from the iliac crest and no other kind of bone grafts were used.

### 3.4. Adverse Events

The overall complication rate was 27% (9/33) with three (9%) medical complications and six (18%) surgery- and implant-associated complications: four complications occurred in the DP group and five in the RP group. Surgery- and implant-associated complications were as follows: Two patients suffered from a transient dysphagia, one each in the DP and the RP group. In one case in which an AO Type C fracture at level C4/5 was treated with a dynamic plate, inadequate reduction was noticed in the postoperative CT scan, which was revised immediately. Moreover, one case with relevant loss-of-reduction by 22° that also showed signs of implant loosening and pseudarthrosis (RP) had to be revised 20.8 months after surgery. This patient received treatment of an AO Type A fracture with a rigid plate and ICBG. Including this case, altogether, in three patients, no fusion could be achieved after a mean radiological follow-up of 19.3 ± 4.5 months and all of them were revised: Two patients with an AO Type A and B fracture treated with a rigid plate and one patient with an AO Type A fracture treated with a dynamic plate. Summarized, revision surgery was necessary in four patients (12%; DP: *n* = 2; RP: *n* = 2).

We observed no differences between the two plate concepts regarding complication rate or revision rate (*p* ≥ 0.05). Furthermore, no differences in pseudarthrosis rates between ICBG and cages could be observed (*p* ≥ 0.05). The patients that received a cage with local bone graft were significantly older than patients that were treated with an ICBG (mean age 64.8 ± 13.6 years vs. 44.6 ± 22.2 years; *p* < 0.05).

### 3.5. Radiological Outcome

#### 3.5.1. Reduction and Loss of Reduction in Terms of mEA and bEA

Monosegmental injuries showed a mean mEA of 11° ± 15° ranging from −17° to 33°, whereas bisegmental injuries ranged from −9° to 20° with a mean bEA of 5° ± 10°.

With the rigid plate system, a reduction of 9° ± 18° with mono- and of 1° ± 2° with bisegmental fixation could be reached (both *p* ≥ 0.05). There was no significant loss of reduction (−1° ± 3°) during follow-up in the mEA (Figure 3A). We observed a loss in the bEA by 8° ± 9° (*p* ≥ 0.05) (Figure 3B). Notably, in this group, one patient had to be revised due to a clinically relevant loss of lordosis by 22°.

With the dynamic plate system, a reduction of 10° ± 13° with mono- and of 1° ± 8° with bisegmental fixation could be reached (both *p* ≥ 0.05). There was no significant loss of reduction by −0° ± 3° during follow-up in the mEA (Figure 3A). We observed a loss in the bEA of 2° ± 5° (*p* ≥ 0.05) (Figure 3B).

#### 3.5.2. Anterior Translation

With both plate systems, the preoperative translation could be significantly reduced when compared to the follow-up (both *p* < 0.01; no significant differences between plate systems). During the follow-up time, we observed a loss of reduction by 0.2 mm (*p* ≥ 0.05) in both groups (Figure 4).

#### 3.5.3. The Anterior Wall and Posterior Elements

In the monosegmental fusions, the height of the anterior wall was significantly restored by 3.0 ± 4.3 mm (*p* < 0.01). There was no significant change in the height of the posterior wall postoperatively (*p* ≥ 0.05). During the follow-up, there was a significant loss in height, both in the anterior (0.6 ± 1.3 mm) and in the posterior wall (0.7 ± 1.1 mm) (both *p* < 0.05). In bisegmental fusions, the height of both the anterior and posterior wall did not decrease during follow-up (both *p* ≥ 0.05). Analyzing the subgroup of the dynamic plate system showed that no significant loss of reduction of the anterior or posterior wall in mono- and bisegmental fusions during follow-up occurred (all *p* ≥ 0.05).

#### 3.5.4. Fusion Rate

Bony consolidation could be observed at the last radiological follow-up in 30/33 patients (91%) with no significant differences between both groups (DP: 12/13 (92%); RP: 18/20 (90%); (*p* ≥ 0.05)). 

### 3.6. Health-Related Quality of Life

The mean EQ VAS reached 72.0 ± 4.9 points and the mean EQ-5D Index was 0.8 ± 0.1 at follow-up. There was no difference in the EQ VAS or Index depending on dynamic or rigid procedures (all *p* ≥ 0.05). Figure 5 displays the subdimensions of EQ-5D. 

The mean NDI of the total cohort was 11.0 ± 9.3 points, indicating mild disability. Again, no significant difference was observed between the plate concepts (all *p* ≥ 0.05). 

Looking at the SF-36, the mean PCS of the whole cohort was 41.9 ± 16.2 and the mean MCS was 45.4 ± 14.9. No statistically significant difference in the PCS and the MCS was detected, comparing the DP (PCS 35.9 ± 17.9; MCS 45.8 ± 17.7) and the RP (PCS 48.8 ± 9.4; MCS 44.8 ± 10.6) group (all *p* ≥ 0.05).

A subgroup analysis of the SF-36 for patients depending on the severity of the total injury was conducted: There were seven severely injured patients with an ISS ≥ 16 that returned the PROM questionnaires. The mean ISS in this group was 36 ± 15 points. In a subgroup analysis, the PROMs of the 13 patients with an ISS < 16 points were evaluated: Table 2 displays the results of the 8 main SF-36 items for the population of patients with ISS < 16 in comparison to the normative data of the German population [15]. 

## 4. Discussion

This study, for the first time, demonstrates that a dynamic plate concept provides adequate stability for traumatic cervical fractures including AO Type B and C fractures. The dynamic plate concept showed a similar performance compared to the conventional rigid plate system in terms of postoperative loss of reduction and PROM. 

The use of anterior cervical locking plates evolved to treat cervical disco-ligamentous injuries and fractures of the cervical spine during the last decade. The procedure has been proven to be safe and efficient [17,18]. The data on the performance of dynamic plate concepts in traumatic cervical fractures, concerning loss of reduction, fusion-rates and PROM were observed.

Pitzen et al. investigated the biomechanical effectiveness of dynamic and rigid cervical plates in a C4–7 spine segments cadaver model [11]. They discovered that with both cervical plate systems, a single segment flexion distraction injury could be stabilized, with the dynamic plate design being at advantage in extension. The dynamic plate transmitted 30% less strain through the plate than the rigid plate did [11]. 

Possible clinical advantages of dynamic plate designs have been shown again by Pitzen et al. in their prospective, controlled, randomized, radiological and clinical follow-up of 132 patients [10]. In the group treated with dynamic plates, no implant complications were documented, whereas implant-associated complications were found in four cases of the control group, including plate-breakage and screw loosening. Interestingly, the speed of fusion was faster using a dynamic plate [10]. However, the indications for routine ACDF in their study were heterogeneous with mostly degenerative disc disease. Only nine patients with cervical fractures were included. In contrast, our study population exclusively consists of trauma patients without known degenerative spinal diseases. We found three incomplete fusions, two in patients treated with the rigid plate system and one in a patient with the dynamic plate concept after a mean radiological follow-up of 19.3 months in those three cases. The overall fusion rate of 91% can be considered satisfactory, without a statistical difference between the groups (DP: 92%; RP 90%). Similarly, Goldberg et al. did not find significant differences in the fusion rates between dynamic (89.0%) and static plates (87.8%) in their short-term review of two-level ACDFs [19].

We could not identify any differences in fusion-, complication rates, loss of reduction or PROM between the usage of an ICBG or a cage. This is in line with the majority of recent studies that report satisfactory radiological and clinical outcomes independent of the usage of a cage or an ICBG [20,21].

In terms of the radiological follow-up, we demonstrated that with both plate systems, stabilization of the anterior column can be performed efficiently in mono- and bisegmental fusions. We observed a relevant decrease in the anterior (0.6 mm) and posterior (0.7 mm) wall. This finding suggests that the loss was due to subsidence.

In a radiographic review of 87 patients with either unilateral or bilateral facet dislocations or fracture/dislocations treated with ACDF, and plating with static plate systems, Johnson et al. observed radiographic failures in 13% of the cases [22]. Failure was defined as a change in translation of greater than 4 mm and/or change in angulation of greater than 11 degrees between the immediate postoperative films and follow-up. If these criteria were applied on our cohort, failure was only present in one case in the rigid plate group with a translation of 4 mm and change in angulation of 21.9° that had to be revised 20.8 months after surgery, when implant loosening was detected. No comparable loss of reduction was observed in the dynamic plate group. Looking at the whole study population, a significant reduction could be reached with ACDF and both systems could secure this reduction over a mean radiological follow-up of 29.8 months. However, a statistically insignificant change as well in the mEA and the bEA was seen in both groups at follow-up. Again, subsidence must be considered. Furthermore, it must be taken into account that measurements of the postoperative CT scan in the lying position were compared to lateral X-rays at follow-up in the standing position.

In line with our findings, Dubois et al. noted no difference in the clinical outcome in their retrospective analysis of 52 patients who underwent two- or three-level ACDF with a rigid or a dynamic plate [23]. They were not able to conclude any clinical or radiographic advantages of the dynamic plate during 20.9 months of follow-up, compared to the rigid plate (follow-up of 13.9 months). However, the authors did not assess objective outcome measures, rather the Odom scale [23]. Similar results regarding the Odom scale were reported by Stancić et al. [24]. 

We showed satisfactory results in terms of the mean EQ-5D VAS of 72.0 ± 4.9 points and a mean NDI of 11.0 ± 9.3 points in the total study cohort without a significant difference between the two groups. Due to the trauma setting, there was no preoperative subjective patient reporting that could be compared to. There were no relevant differences in PROM between the two study groups. For these reasons, our results suggest that dynamic and rigid cervical plating is a suitable method to achieve an adequate stabilization and aim for a solid fusion and good quality of life after traumatic injury of the cervical spine. Unlike other studies, we showed that an important factor for good PROM was the ISS: In our cohort, the less severely injured patients with an ISS < 16 reached the level of a healthy German population in terms of the SF-36 in the mid-term follow-up [15].

The main limitation of the current study is the cohort size of 33 patients. Therefore, the statistical analysis must be interpreted with caution. However, our cohort consists exclusively of trauma patients with mostly AO Type B or C injuries (79%) of the cervical spine. There was no significant difference in fracture type distribution between the two groups. Different from other studies, no patients with degenerative spinal diseases were included. By providing homogeneity in the treatment of patients and in the data analysis as well as by the above presented subgroup analysis, we tried to minimize eventual bias effects.

It is controversially discussed if anterior plate fixation for AO-Type C injuries would require additional posterior fixation, especially in the cervicothoracic junction. Notably, closed or open reduction via the anterior approach could be achieved in all patients of the current cohort. Although complex facet fracture dislocations were not an exclusion criterion, this might represent a potential inclusion bias, as patients with higher grades of instability possibly have been treated with a combined anterior and posterior approach to ensure sufficient reduction and stability. In our cohort, we documented six fusions that affected the cervicothoracic junction in C7 fractures (*n* = 3 DP; *n* = 3 RP) from which only one presented a pseudarthrosis at follow-up and had to be revised (DP). We did not find relevant fracture dislocation at the cervicothoracic junction after treatment with a dynamic plate.

## 5. Conclusions

An initial traumatic instability or anterolisthesis of subaxial cervical spine fractures could be reduced and stabilized using both the dynamic and the rigid plate. The dynamic plate provided enough stability in a fragile fracture situation with no relevant loss of reduction during follow-up. In terms of fusion rate and complications, the dynamic plate system was not inferior to the rigid plate concept. Overall PROM was satisfactory in both groups. Patient-reported outcome in terms of the SF-36 was similar to a healthy reference population in non-polytraumatized patients (ISS <16) with subaxial cervical spine fractures in mid-term follow-up.

## Figures and Tables

**Figure 1 jcm-10-01185-f001:**
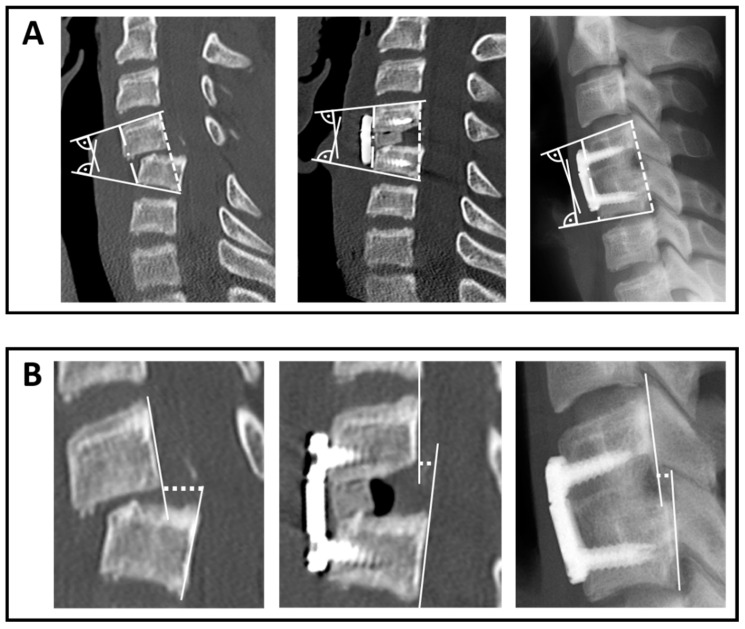
Example CT scan pre- and postoperative and lateral X-ray scan at follow-up. (**A**) Measurement of the mEA and the ventral and dorsal wall. The mEA (or bEA) was measured between the upper endplate of the superior vertebra and the inferior endplate of the inferior vertebra of the affected segment. The dotted lines indicate the measurement of the ventral and dorsal wall. (**B**) Measurement of the sagittal translation. The dotted lines mark the measurement of the sagittal translation.

**Figure 2 jcm-10-01185-f002:**
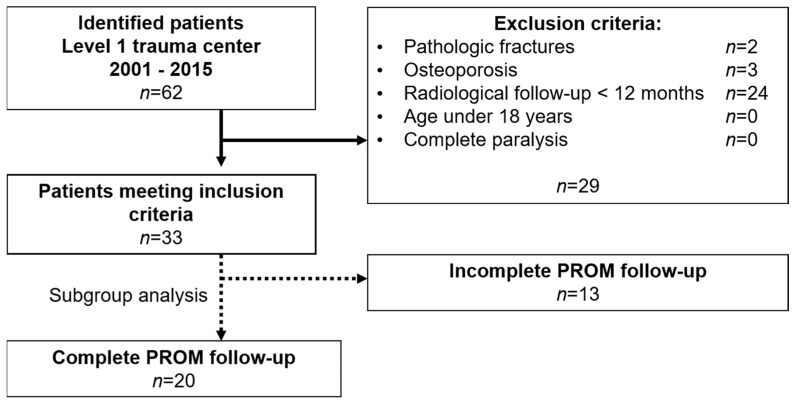
Flowchart of patient enrollment. Sixty-two patients with subaxial cervical fractures were identified between 2001 and 2015. After applying the exclusion criteria, 33 patients were eligible to be included in the study cohort. Twenty patients completed the PROM follow-up, at least 24 months after surgery.

**Figure 3 jcm-10-01185-f003:**
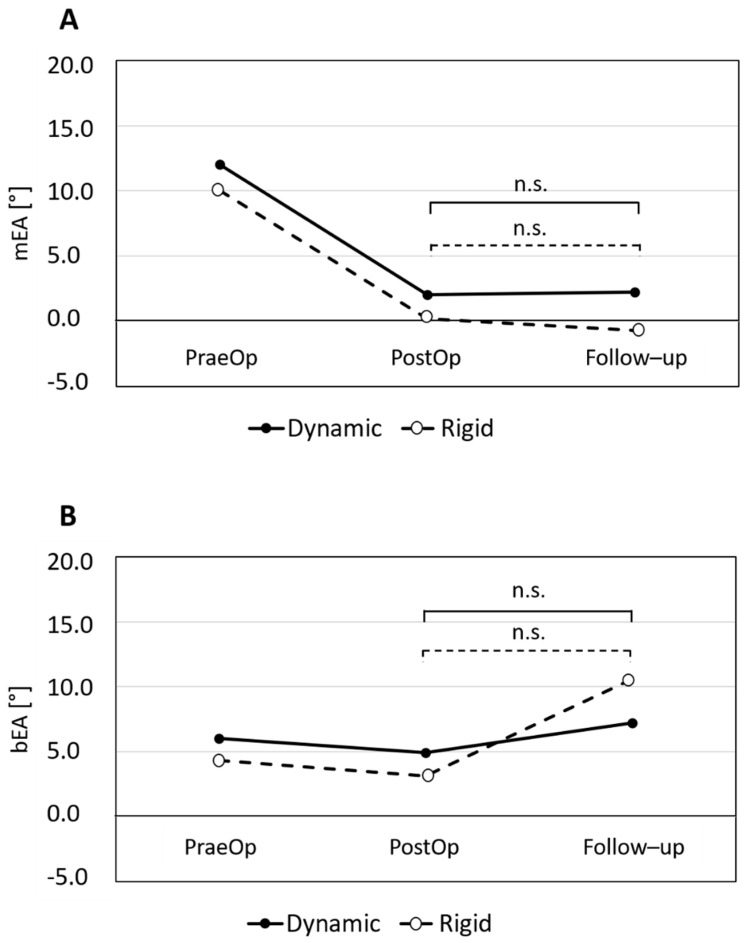
Changes in the mEA (**A**) and bEA (**B**) in the dynamic and rigid plate system. There was no significant loss of reduction during the follow up in both groups (A and B).

**Figure 4 jcm-10-01185-f004:**
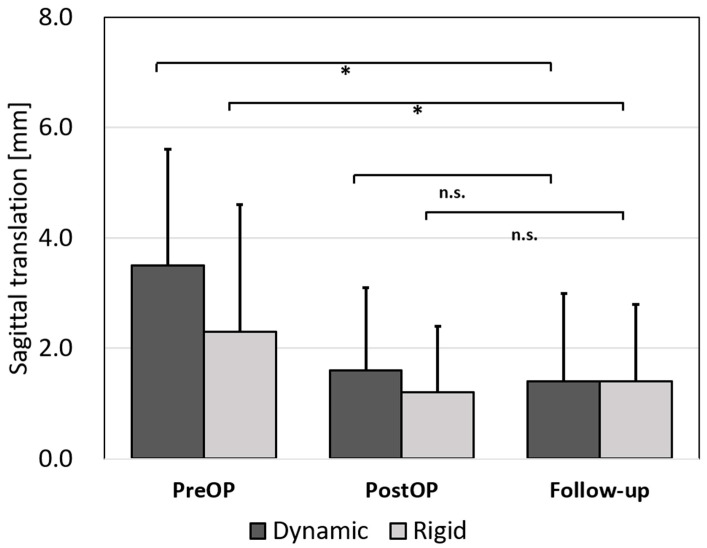
Changes in the sagittal translation. Sagittal translation pre-, postoperative and at the time of the last radiological follow-up. Bars indicate mean values and whiskers the SD. * indicates *p* < 0.05.

**Figure 5 jcm-10-01185-f005:**
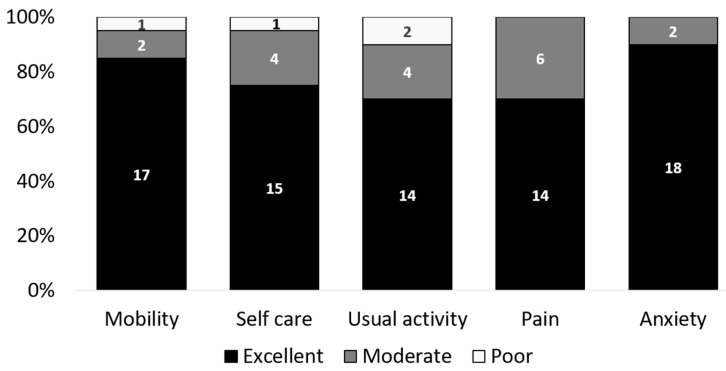
Subdimension of EQ-5D. Categories differ between excellent, moderate, and poor results. The bars represent 100% of the included cases (*n* = 20).

**Table 1 jcm-10-01185-t001:** Distribution of patient data, surgery information and fracture classification.

	Dynamic Plate Design *n* = 13	Rigid Plate Design *n* = 20	*p*
Mean age [years]	47.9 ± 24.5	48.5 ± 21.3	0.95
Female	*n* = 4 (30.8%)	*n* = 4 (20.0%)	
Male	*n* = 9 (69.2%)	*n* = 16 (80.0%)	0.49
Polytrauma	*n* = 6 (46.2%)	*n* = 8 (40.0%)	0.73
ICBG	*n* = 10 (76.9%)	*n* = 17 (85.0%)	0.71
Cage	*n* = 3 (23.1%)	*n* = 3 (15.0%)	
AO Type A	*n* = 4 (30.8%)	*n* = 3 (15.0%)	0.64
AO Type B	*n* = 2 (15.4%)	*n* = 5 (25.0%)	
AO Type C	*n* = 7 (53.8%)	*n* = 12 (60.0%)	

No statistically significant differences were observed for mean age, sex, polytrauma, ICBG/cage, and AO Classification between the study group and the control group.

**Table 2 jcm-10-01185-t002:** The eight main SF-36 items results of a population with ISS < 16 in comparison to the normative data of the German population [15].

SF-36 Item	German Reference Population 2013 [15]	Study Population ISS < 16 (*n* = 13)	*p*
Physical functioning	89.5 (88.3–90.7)	75.0 ± 33.8	0.166
Role physical	85.5 (84.1–86.9)	68.8 ± 37.1	0.146
Role emotional	86.8 (85.3–88.2)	69.4 ± 38.8	0.150
Vitality	60.7 (59.4–61.9)	55.8 ± 20.5	0.429
Emotional Well-Being	72.8 (71.6–73.9)	74.7 ± 18.2	0.730
Social functioning	85.6 (84.2–87.1)	82.3 ± 27.4	0.684
Bodily Pain	75.3 (73.6–76.9)	76.1 ± 23.4	0.910
General health	69.9 (68.8–71.1)	67.0 ± 19.3	0.608

The reference values are presented as mean with 95% confidence interval. There is no significant difference between the study group (ISS < 16) and a healthy German population in all main items.

## Data Availability

All data presented in this study are available on demand from the corresponding author.

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
