# Peer review of "Does Dynamic Anterior Plate Fixation Provide Adequate Stability for Traumatic Subaxial Cervical Spine Fractures at Mid-Term Follow-Up?"

_jcm, 2021, doi:10.3390/jcm10061185_

Round 1

Reviewer 1 Report

Congratulations for this interesting manuscript. The topic is surely of interest and  the paper is nicley written. Nevertheless I suggest some adaoptions before publication:

Radiographic assessment

What do you mean by anterior/posterior column? Please specify which classification you use. In the x-ray you show the anterior and posterior wall, not column. The Denis classification uses 3 columns, the AO 2. Both classifications don`t correlate with your picture.

Trauma Mechanism and Fracture Classification

19 patients were treated due to type c injuries to the cervical spine. Some would argue that anterior plate fixation in this type of injury would require additional posterior fixation, especially in the cervicothoracic junction. Could you please comment on that?

Treatment strategy and complications

Why some patients received ICBG and some a titanium cage? Was it due to surgeon’s preference alone or are there other factors?

 Adverse events

Please specify in which group and at what kind of injury the adverse events took place. The severe pseudarthrosis you are talking about, was it in the dynamic or rigid group. What was the initial classification of the injury. Please specify that for all complications.

 Fusion rate

How did you evaluate fusion without a CT scan? Please specify your criteria.

 Mono vs bi vs trisegmental stabilization

You mentioned 1,2 and 3 level ACDF`s. Later on you just talk about 1 and 2 levels. Please either be consistent.

Author Response

Dear reviewer,

We would like to thank you for your valuable and structured comments and suggestions, helping to improve our manuscript. In the following we would like to answer your questions as well as point out the changes we made.

If any further questions or any dubiety appears, please do not hesitate to contact us.

Reviewer 2 Report

I have two concerns.  The first is that your follow-up with PROMS was a very small group of participants.  This can lead to some questions about representativeness.  Were the participants with good outcomes more likely to respond, or complete the PROMS.  The second concern is that using a lack of statistical significance with such a small group is not the same as testing equivalence of two groups.  You mention non-inferiority but that is not truly what you tested.  Given the small group that you did have complete data on the chance of finding statistically significant results is low, even if there was significant difference between the groups.  

For instance table 2.  first the header of the final column needs to be changed to something other than significant differences, since you are claiming none.  But also examination of the SF-36 scores for your participants escapes the 95% CI on every one of the 8 main items ( i am assuming subscales).  However, the wide variability in your small sample makes it see as if your sample is not different than the population norms.  This may be misleading. there is also a typo in table 2 in the German reference column physical functioning mean is outside the 95% CI reported for the range.  Not sure how this is possible.   

Another concern is the use of the significance of the Pearson Correlation.  this is a meaningless descriptive statistic and has no purpose in inferential setting.  

Author Response

Dear reviewer,

We would like to thank you for your valuable comments and suggestions. In the following we would like to answer your questions as well as point out the changes we made.

If any further questions or any dubiety appears, please do not hesitate to contact us.
